# A Dual and Rapid RPA-CRISPR/Cas12a Method for Simultaneous Detection of Cattle and Soybean-Derived Adulteration in Goat Milk Powder

**DOI:** 10.3390/foods13111637

**Published:** 2024-05-24

**Authors:** Yuanjun Wen, Shuqin Huang, Hongtao Lei, Xiangmei Li, Xing Shen

**Affiliations:** Guangdong Provincial Key Laboratory of Food Quality and Safety, College of Food Science, South China Agricultural University, Guangzhou 510642, China; 17820593218@163.com (Y.W.); hq_down_0309@163.com (S.H.); immunoassay@126.com (H.L.); lixiangmei12@scau.edu.cn (X.L.)

**Keywords:** goat milk powder, CRISPR/Cas12a, adulteration, simultaneous detection, food fraud

## Abstract

The adulteration of goat milk powder occurs frequently; cattle-derived and soybean-derived ingredients are common adulterants in goat milk powder. However, simultaneously and rapidly detecting cattle-derived and soybean-derived components is still a challenge. An efficient, high-throughput screening method for adulteration detection is needed. In this study, a rapid method was developed to detect the adulteration of common cattle-derived and soybean-derived components simultaneously in goat milk powder by combining the CRISPR/Cas12a system with recombinant polymerase amplification (RPA). A dual DNA extraction method was employed. Primers and crRNA for dual detection were designed and screened, and a series of condition optimizations were carried out in this experiment. The optimized assay rapidly detected cattle-derived and soybean-derived components in 40 min. The detection limits of both cattle-derived and soybean-derived components were 1% (*w*/*w*) for the mixed adulteration models. The established method was applied to a blind survey of 55 commercially available goat milk powder products. The results revealed that 36.36% of the samples contained cattle-derived or soybean-derived ingredients, which revealed the noticeable adulteration situation in the goat milk powder market. This study realized a fast flow of dual extraction, dual amplification, and dual detection of cattle-derived and soybean-derived components in goat milk powder for the first time. The method developed can be used for high-throughput and high-efficiency on-site primary screening of goat milk powder adulterants, and provides a technical reference for combating food adulteration.

## 1. Introduction

Since 1960s, goat milk products have received widespread attention. Goat milk powder has the largest share of the market for goat milk products in China. It is preferred by consumers due to its high nutritive value [1,2] and low allergenicity [3]. Recently, the market for goat milk products has been expanding. However, due to the scarcity of goat milk resources and high production costs, adulteration is a common problem [4,5]. The most common way of adulteration is through the use of inexpensive and readily available cow’s milk [6]. Soybean milk is another common adulterant. Such adulteration not only harms consumers’ rights and interests, but also causes problems such as food allergies. The proliferation of adulteration has attracted attention [7], and various methods have been developed to detect it.

Analytical techniques such as electrophoresis [8], spectral analysis [9], mass spectrometry [10], and fingerprinting techniques [11] have been used to analyze the composition of goat milk products. These techniques require expensive equipment and intricate operation. Immunochromatographic test strips have become a commonly used method for goat milk adulteration detection in market supervision nowadays due to the advantages of easy portability, simple operation, and fast response. However, due to the easy denaturation or cleavage of proteins during the processing of dairy products, the performance of protein-based detection methods is unstable. The protein level detection method is often only applicable to raw goat milk or lightly processed dairy products, and cannot be adapted to a large number of dairy products including milk powder.

Nucleic acid detection assay has become the mainstream method of species identification nowadays because of its advantages of high sensitivity and specificity [12,13]. DNA-based methods for the detection of adulteration in goat milk include traditional PCR [14], the DNA-based fluorometric microspheres method [15], high-resolution melting (HRM) [16], and real-time PCR [17,18]. Such methods require precision instruments and specialized technicians. The speed and convenience of isothermal detection methods such as loop-mediated isothermal amplification (LAMP) enable them to simplify the detection process while maintaining high sensitivity, realizing rapid on-site detection of goat milk adulteration [19,20], but LAMP uses multiple pairs of primers, making the system more unstable.

CRISPR/Cas, a revolutionary method for gene editing, has emerged as a rising diagnostic tool [21]. Compared with the existing nucleic acid detection techniques, this method has great potential in developing rapid and accurate on-site detection techniques due to its simplicity of operation, high sensitivity and specificity, and low reliance on precision instruments [22]. CRISPR/Cas detection technology has been applied in the field of medical diagnosis, and has also been gradually applied to the rapid detection of food safety, including genetically modified crops [23], meat adulteration [24], the detection of foodborne pathogens [25,26], and so on. Previously, we established a method to detect adulteration of goat milk powder with cattle-derived components based on CRISPR/Cas12a [27]. However, a detailed analysis of the products available on the market revealed that soybean-derived components are also common adulterants in goat milk powder. Cattle-derived components and soybean-derived components belong to two different sources, animal and plant respectively, makes simultaneous nucleic acid co-extraction challenging, which leads to difficulties in rapid simultaneous detection. At present, there is little research on dual detection of adulteration in milk products. Since milk powder contains very little nucleic acid and a large amount of protein, polysaccharide and other impurities, the difficulty of simultaneous detection further increases.

In this study, in order to meet the demand for simultaneously detecting cattle-derived components and soybean-derived components in goat milk powder, a dual detection method was established using CRISPR/Cas12a coupled with recombinant polymerase amplification (RPA) technology. The specificity, sensitivity, and validity of the method were fully evaluated, and the feasibility of the method was demonstrated by blind examination of commercial samples. In this study, DNA from cattle-derived and soybean-derived ingredients was extracted simultaneously at the same concentration level. The establishment of the dual RPA-CRISPR/Cas12a method realizes the simultaneous extraction, amplification, and detection of animal and plant DNA for the first time, which enables high-throughput and high-efficiency screening of goat milk powder adulterants, and provides a technical reference for food adulteration detection. The blind survey of 55 goat milk powder products showed that 36.36% of the samples contained cattle-derived or soybean-derived ingredients.

## 2. Materials and Methods

### 2.1. Materials

All primers, ssDNA, and crRNA were purchased from Genewiz Biotechnology (Suzhou, China). An RPA assay kit (TwistDW, Cambridge, UK), DNA isothermal rapid amplification kit (Amp-Future Biotech, Changzhou, China), Cas12a enzyme (Editgene, Guangzhou, China), RNase inhibitor (Hai Gene, Harbin, China), NEBuffer 2.1 (New England Biolabs Inc., Ipswich, UK), DreamTaq DNA Polymerase (Thermo Fisher Scientific, Waltham, MA, USA), qPCR Probe Master Mix (Vazyme Biotech, Nanjing, China), and cattle milk powder (Yili company, Hohhot, China) were used for the extraction, amplification, and detection of animal and plant DNA. The standard goat milk powder was acquired from Hongxing Meiing Dairy (Weinan, China). The soybean milk powder was sourced from Wuzhou Bingquan Industry (Wuzhou, China). A total of 55 whole goat milk powder products was purchased from Chinese online shopping platforms TaoBao, Pinduoduo, and TikTok. The flowchart of the methodology used in this paper is shown below (Figure 1).

### 2.2. DNA Extraction

Usually, it is difficult to co-extract nucleic acids from both animal and plant sources. In this study, cattle milk powder and soybean milk powder were respectively mixed into goat milk powder at different mass ratios, and the genomic DNA was extracted by the same method, developed from an alkaline decomposition method [28], realizing the simultaneous extraction of animal- and plant-derived nucleic acids. Four grams of milk powder was dissolved in 40 mL of saturated NaCl solution and centrifuged at 4500 rpm for 10 min at 4 °C. The precipitate was resuspended with 1 mL ddH_2_O and 200 μL alkaline lysis solution (0.5 mol/L NaOH, 0.1 mol/L Na_2_EDTA). The solution was placed in a boiling water bath for 5 min and then centrifuged at 12,000 rpm for 5 min. To the supernatant was added 2.5 times the volume of anhydrous ethanol and it was shaken well for 1 min. The tubes were kept at 4 °C for 30 min. The solution was centrifuged at 8000 rpm for 10 min at 4°C, and the supernatant was discarded. The precipitate was resuspended in 1 mL of 70% ethanol, centrifuged at 4 °C and 12,000 rpm for 5 min, and the supernatant was again discarded. The precipitate was allowed to dry at room temperature for 5 min and resuspended with 50 µL of sterilized ultrapure water. Purified DNA was measured for concentration using a NanoDrop 2000c (Thermo Fisher Scientific, Waltham, MA, USA) and stored at −20 °C for later use.

### 2.3. Primer Screening

According to the standard method described in the Technical Regulations for Authenticity Identification of Goat Milk (NY/T 3050–2016), the endogenous gene of soybean lectin (NM_001354824.1) and cattle mitochondrial DNA (NC_006853.1) were chosen as the detection targets. RPA primers were designed using Primer Premier 5.0 software following the instructions of TwistDx^®^, and were initially screened with the Primer-BLAST tool on the NCBI website (https://www.ncbi.nlm.nih.gov/tools/primer-blast/ (accessed on 6 September 2021)). Eight pairs of primers for cattle and soybean were designed, respectively. The cattle primer pairs and soybean primer pairs were mixed in a 1:1 ratio and the genomic DNA extracted from pure cattle milk powder and soybean milk powder were used as templates for primer screening, according to the RPA reaction system recommended in the instruction manual of the commercial amplification kit. The RPA amplification products were analyzed by agarose gel electrophoresis to screen out the primer pairs with clear and specific amplification bands. Subsequently, the DNA of cattle, goat, and soybean were respectively utilized to detect their interspecies specificity of the primers screened out.

At the same time, the primers were also examined by a primary CRISPR/Cas12a system, on the basis of the instruction manual for the Cas12a enzyme. Cattle crRNA and soybean crRNA were mixed at the ratio of 1:1. The RPA products were added into the dual CRISPR/Cas12a system and incubated for 30 min. The primer combinations with good specificity and bright fluorescence observed by the naked eye were finally selected.

### 2.4. RPA Optimization

In order to improve the reaction efficiency and excite fluorescence signals with the same intensity from the cattle and soybean genomic DNA, the RPA system was optimized. The optimization parameters included primer concentrations of each pair (300 nM, 400 nM, 500 nM, 600 nM), primer ratios (4:3, 3:2, 2:1, 1:1, 1:2, 2:3, 3:4), dNTP concentrations (0.8 mM, 1.2 mM, 1.6 mM, 1.8 mM, 2 mM, 2.4 mM), MgOAc concentrations (7 mM, 10.5 mM, 14 mM, 17.5 mM, 21 mM), and amplification time (5 min, 10 min, 15 min, 20 min, 25 min, 30 min). The reaction products were analyzed in the light of fluorescence intensity provided by the initial dual CRISPR/Cas12a system.

### 2.5. Establishment of Dual RPA-CRISPR/Cas12a Assay

In this study, for crRNA corresponding to cattle-derived and soybean-derived targets, we designed the online tool Cas-Designer (http://www.rgenome.net/cas-designer/ (accessed on 1 December 2021)), and preliminarily screened for their specificity using the BLAST tool on the NCBI website (https://www.ncbi.nlm.nih.gov/tools/primer-blast/ (accessed on 1 December 2021)). The settled sequences were 5′-UAAUUUCUACUAAGUGUAGAUCACAAUCCAGAACUGACAC-3′ for cattle crRNA, and 5′-UAAUUUCUACUAAGUGUAGAUCCCAAAUGUGGAUGGGGGGGU-3′ for soybean crRNA. Then, the reaction conditions of the CRISPR system were also optimized to achieve equivalent detection efficiency for cattle-derived and soybean-derived ingredients. First, the ratio of cattle crRNA and soybean crRNA was set at 1:1, and a range of Cas12a enzyme concentrations of 50 nM, 75 nM, 100 nM, 125 nM, and 150 nM were tested. Based on the optimal Cas12a enzyme concentration, different ratios of total crRNA to Cas12a enzyme, including 0.5:1, 1:1, 1.5:1, 2:1, and 2.5:1 were optimized. Then, the optimal cattle and soybean crRNA ratios in total crRNA were investigated at 2:1, 3:2, 8:7, 1:1, 7: 8, 2:3, and 1:2.

To carry out the experimental operation, the reaction components of the RPA amplification system were blended in the tube, and the MgOAc was added to the cap of the tube. The reaction was started by mixing the MgOAC into the RPA tube after a slight centrifugation and incubated at 37 °C for 20 min. At the end of the RPA reaction, 4 µL of RPA products was added to the CRISPR system, then it was mixed uniformly, and kept at 37 °C. When cattle-derived or soybean-derived components were present in the samples, the CRIPSR system produced green fluorescence visible to the naked eye under a mini BluView Transilluminator (Eastwin Life Sciences, Inc., Beijing, China) at the wavelength of 470 nm. The fluorescence intensity was also immediately captured via QuantStudio 3 Real-Time PCR (Thermo Fisher Scientific, Waltham, MA, USA). The no-treatment control (NTC) employed ddH_2_O as a template instead of a DNA template.

### 2.6. Specificity and Detectability of Dual RPA-CRISPR/Cas12a

We validated the specificity of the established dual RPA-CRISPR/Cas12a method for the detection of cattle and soybean-derived components using the DNA from cattle, goat, and soybean as templates. The absolute detectability of this approach was confirmed on diluted cattle and soybean genomic DNA with a concentration gradient ranging from 10^−6^ ng/µL to 100 ng/µL. Meanwhile, a battery of goat milk powder models adulterated with different proportions (50%, 20%, 10%, 5%, 1%, and 0.1% (*w*/*w*)) of cow’s milk powder or soybean milk power, respectively, was used to assess the detection limit of the dual RPA-CRISPR/Cas12a method.

### 2.7. Real Sample Testing and Statistical Analysis

To validate the utility of the established dual RPA-CRISPR/Cas12a approach, 55 different goat milk powder products were evaluated by the dual RPA-CRISPR/Cas12a approach. Each assay was repeated three times. When the green fluorescence was observed, it indicated the presence of cattle or soybean-derived components in the sample, which was judged as positive. In contrast, no green fluorescence indicated a negative result.

Meanwhile, the fluorescence intensity was collected by real-time PCR and analyzed by SPSS 21.0 multivariate analysis software (IBM, Armonk, NY, USA). The comparison between the experimental group and the control group was performed by unpaired two-tailed t-test, and the significance of differences was expressed as a *p*-value. A statistically significant difference at *p* < 0.05 (*) was considered a valid experiment.

### 2.8. Method Validation

To validate the accuracy of the dual RPA-CRISPR/Cas12a assay, 55 goat milk powder product samples were analyzed by real-time PCR concurrently. The accuracy of cattle-derived ingredient detection was validated according to a Chinese Group Standard “Qualitative detection method of cattle (domestic cattle, yak and buffalo) and sheep (goat and sheep)-derived components in milk and dairy products (T/CNHFA 002-2022, China)”. When the CT value was ≤30, the samples were judged to be positive. When the CT value was ≥35, the samples were judged to be negative. When the value was <30 to <35, the experiment was repeated, and the samples were judged positive if the CT value < 35. The accuracy of soybean-derived ingredient detection was validated following a Chinese National Standard, “PCR method for the characterization of standard genes within soybean for the detection of the composition of genetically modified plants and their products”. When the CT value was ≤36, the sample was judged as positive. When the value was <36 to <40, the experiment was repeated. If the result of repeated experiments was still ≤36 to ≤40, it was judged as positive. Pure cattle DNA and soybean DNA were used as positive controls. Pure goat DNA and ddH_2_O were employed as negative and blank controls, respectively. The experiment was valid when both the internal reference gene test and positive control were positive, and the negative control and blank control were negative. The experimental results were analyzed, plotted, and discriminated using QuantStudio^TM^ 3 Design and Analysis software.

## 3. Results and Discussion

### 3.1. DNA Extraction

The nucleic acid extraction method for milk powder samples still suffers from the issues of cumbersome operation and the inability to simultaneously extract DNA from both animals and plants. First, the consistency of nucleic acid levels for both species cannot be guaranteed, which may adversely affect subsequent detection. Second, in contrast to simple tissue from animals and plants, dairy products usually have low nucleic acid content and high levels of impurities that increase the difficulty of extraction. Therefore, in this study, we improved the extraction method based on the common alkaline lysis method. During the sample pretreatment, milk powder was dissolved using a saturated NaCl solution and centrifuged to remove a large amount of impurities. The high salt content extracted a lot of milk proteins from the solution. After the alkaline lysis process, the DNA was further purified using 70% ethanol, forming flocculent precipitates. This method selectively removed most impurities from the milk powder through a simple one-step operation, thereby simplifying the cumbersome sample pretreatment steps and reducing the loss of DNA. The resulting DNA levels after simultaneous extraction were similar, with a final concentration of 96 ng/µL for cattle DNA and 115 ng/µL for soybean DNA.

### 3.2. Primer Screening and RPA Optimization

RPA amplification primer combinations were screened from four cattle primer pairs and three soybean primer pairs. One combination containing two pairs of primers could meet the need for efficient amplification of cattle- and soybean-derived components at the same time (Appendix A). The specificity of primer combinations was examined among cattle, goat, and soybean and the results showed good specificity (Figure 2). It showed that both cattle and soybean were able to emit green fluorescence, while no fluorescence was produced in goat primers. The final cattle forward primer was 5′-GGGTTACGAGAGGGAGACCTAAAATTACAG-3′ and the reverse primer was 5′-GCTTGGGAATAGTACGATGCCGCAACTAGA-3′. The soybean forward primer was 5′-ACGCTATTGTGACCTCCTCGGGAAAGTTAC-3′ and the reverse primer was 5′-AGATAACCTGCATGTGTTTGTGGGCTTAGTG-3′.

The appropriate concentration was first explored separately for cattle and soybean primers. The results showed that the amplification efficiency was highest for both targets when the primer concentration used for cattle or soybean was 600 nM (Appendix A). In the dual RPA amplification, two primer pairs compete with each other for important components such as enzymes, dNTPs, and MgOAc, and one often inhibits the other. In order to ensure that the amplification efficiency can be consistent for both components in the same system, the ratio between cattle and soybean primers was optimized. The amplification efficiency of the two targets was basically the same when the primer ratio between cattle and soybean was 3:4 (Appendix A).

Other factors, including dNTPs, MgOAc, and amplification time, were also investigated. The amplification efficiency fluctuated slightly with the change of dNTPs concentration (Appendix A), and the optimal concentration was 2 mM for both components. MgOAc provides energy to initiate the reaction process in RPA amplification. The reaction efficiency increased gradually with the increase of MgOAc concentration for cattle components (Appendix A), while it varied indeterminately for soybean components (Appendix A). When the MgOAc concentration was 17.5 mM, the amplification efficiency for both targets was at a high level. As the RPA amplification proceeded, the amplification products continued to accumulate. However, the fluorescence intensity gradually decreased with time in the later stages, which may have been due to the quenching of the probe caused by the long reaction time (Appendix A). When the RPA amplification time was 20 min, the dual system reached a high level of RPA amplification for both targets. The final reaction mixture contained 1200 nM primers in total (cattle: soybean = 3:4), 2 mM dNTPs, 17.5 mM MgOAc, 2 µL DNA template, 10 µL 2× reaction buffer, 2 µL 10× basic E-mix, and 1 µL 20× core reaction mix. The reaction was conducted at 37 °C for 20 min.

### 3.3. Establishment of Dual RPA-CRISPR/Cas12a Method

In the CRISPR/Cas12a system, Cas12a, crRNA, and target DNA form a ternary complex to activate the reaction. The concentration of Cas12a directly influences the reaction rate. As the Cas12a enzyme concentration increased, the fluorescence intensity first increased and then decreased. Maximum intensity was achieved at 100 nM and 75 nM Cas12a concentration for cattle-derived and soybean-derived components, respectively (Figure 3a,b). After a comprehensive comparison of the fluorescence intensity, the final Cas12a concentration of 100 nM was selected.

Based on the optimal concentration of Cas12a, the crRNA/Cas12a ratio was investigated. The effects of crRNA/Cas12a ratios were different for cattle and soybean targets (Figure 3c,d). This may have been related to crRNA design. Different crRNA has different efficiency and specificity to bind Cas12a, as well as paring with target DNA. For the consideration of balancing the reaction efficiency, the final crRNA/Cas12a ratio of 1.5:1 was chosen. With two crRNAs in the dual system, changing the ratio between the two crRNAs will also change the ratio of a single crRNA to that of the Cas12a enzyme, which will simultaneously affect the efficiency of the CRISPR reaction and the consistency of the detection results of cattle- and soybean-derived components. In this study, when the assay was performed on cattle-derived components, the reaction efficiency declined generally as the cattle crRNA percentage decreased. However, when the soybean-derived components were detected, the relationship between reaction efficiency and soybean crRNA percentage showed a weak correlation. In the end, the ratio of cattle crRNA to soybean crRNA was selected as 7:8 to keep good consistency in the detection (Figure 3e). The optimal Cas12a enzyme reaction system was composed of 100 nM Cas12a, 70 nM cattle crRNA, 80 nM soybean crRNA, 500 nM reporter gene ssDNA, 0.8 U RNase inhibitor, 2.5 µL of 10× NEBuffer 2.1, 4 µL of DNA template, and was supplemented with sterile water to a total volume of 20 µL. The optimal dual CRISPR/Cas12a system generated sufficient visible green fluorescence under the blue light within 20 min. Thus, the entire dual RPA-CRISPR/Cas12a detection could be accomplished in 40 min at 37 °C.

### 3.4. The Specificity and Sensitivity of Dual RPA-CRISPR/Cas12a Method

DNA from cattle, goat, and soybean were selected as templates to verify the specificity of the dual RPA-CRISPR/Cas12a method. The experimental findings demonstrated that cattle or soybean DNA was able to activate the Cas12a enzyme and emit visible fluorescence (Figure 4a), whereas there was no significant difference between goat and the blank control. This indicated that the dual RPA-CRISPR/Cas12a approach established in this study could differentiate cattle-derived and soybean-derived components from goat.

The detectability of the dual RPA-CRISPR/Cas12a method was assessed at two levels. First, absolute sensitivity was tested at genomic DNA level. When the concentration of both cattle and soybean genomic DNA was ≥10^−2^ ng/μL, green fluorescence could be observed by the naked eye, which was in accordance with the results of statistical analysis (*p* < 0.001 for cattle DNA and *p* < 0.0001 for soybean). When the DNA concentration was <10^−2^ ng/μL, the fluorescence generated was not significantly different from the blank group. The dual RPA-CRISPR/Cas12a rapid assay had the identical extent of sensitivity for the detection of cattle- and soybean-derived components (Figure 4b,c). On the other hand, different proportions of cattle milk or soybean milk powder were mixed with goat milk powder to determine the detection limit of the adulteration level. From the fluorescence value analysis, the detection limits of both cattle and soybean were at 1%. To the naked eye, significant fluorescence produced by cattle and soybean ingredients was observed at 5% adulteration ratio. In general, real adulteration is not low for the pursuit of economic gain. Therefore, naked-eye observation will be enough for practical detection (Figure 4d,e).

### 3.5. Real Sample Testing

Fifty-five commercially available goat milk powder products were purchased to validate the feasibility of dual RPA-CRISPR/Cas12a rapid assay. The results revealed that 17 samples (No. 2, No. 3, No. 4, No. 18, No. 19, No. 20, No. 38, No. 52, No. 54, No. 10, No. 11, No. 14, No. 27, No. 43, No. 44, No. 45, and No. 46) produced green fluorescence visible to the naked eye and these were judged positive samples, consistent with the statistical analysis of the data (Figure 5b). Samples No. 12, No. 50, and No. 51 were also judged positive based on fluorescence values that were significantly different from those of the NTC (Figure 5a,b). The test results showed that among 55 goat milk powder products on the market, a total of 20 positive samples were adulterated, accounting for 36.36% of the total samples. The judgment of positive samples indicated adulteration, whether or not adulteration was judged by the RFU value. In other reports in China, 2–3 out of 10 goat milk powder samples had cattle-derived ingredients [29,30], while our results also included soybean-derived ingredients. Thus, our findings indicate that adulteration in the goat milk powder market is still happening frequently, which should be noted by the regulators.

### 3.6. Method Validation

The method validating adulteration of cattle-derived components in goat milk powder was carried out based on the qPCR method in the group standard (T/CNHFA 002-2022) [27]. A Chinese National Standard “PCR method for the characterization of standard genes within soybean for the detection of the composition of genetically modified plants and their products” was used to validate the detection results of soybean-derived components. The results showed that 20 out of 55 samples of goat milk powder were adulterated. Compared to the dual RPA-CRISPR/Cas12a detection approach, the results were consistent. Among the positive samples, 13 samples were adulterated using only cattle-derived ingredients [27], 5 samples were adulterated with only soy-derived ingredients, and 2 samples were adulterated with both cattle- and soybean-derived ingredients. No. 12, No. 50, and No. 51 samples, which showed very weak fluorescence in the dual RPA-CRISPR/Cas assay, were all adulterated with soybean-derived ingredients. Since the detection limit of the standard method is 0.5 g/kg (equal 0.5‰), which is higher than the limit of this dual detection method (1%), this might have been due to the low content of soybean-derived ingredients present in the samples, which made the fluorescence indistinguishable. In conclusion, the dual detection method has good accuracy in detecting adulterants in goat milk powder.

## 4. Conclusions

Cattle and soybean-derived ingredients are common adulterants in goat milk powder. However, due to the immaturity of universal DNA extraction methods for animals and plants in dairy products, current research on dual detection methods still extracts DNA separately and relies on large precision instruments in the laboratory [31]. In this study, a rapid method was developed to simultaneously detect the adulteration of common cattle and soybean-derived components in goat milk powder by combining the CRISPR/Cas12a system with recombinant polymerase amplification (RPA). This experiment employed a dual DNA extraction method, designed and screened dual detection primers and crRNA, and performed a series of condition optimizations. The optimized assay rapidly detected cattle and soybean-derived components in 40 min. For the mixed adulteration models, the detection limits for both cattle and soybean-derived components were 1% (*w*/*w*). The blind survey of 55 goat milk powder products showed that 36.36% of the samples containing ingredients of cattle or soybean origin. This study combined improved sample pretreatment and alkaline lysis methods to extract DNA from cattle and soybean-derived ingredients simultaneously, thus realizing dual extraction, dual amplification, and dual detection of cattle and soybean-derived components in goat milk powder for the first time, which is very promising for rapid on-site screening.

## Figures and Tables

**Figure 1 foods-13-01637-f001:**
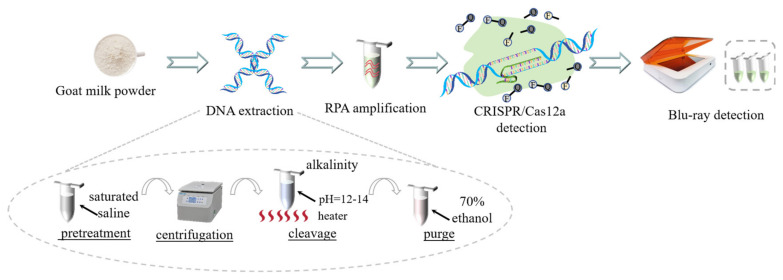
A dual and rapid RPA-CRISPR/Cas12a method for simultaneous detection of cattle- and soybean-derived adulteration in goat milk powder.

**Figure 2 foods-13-01637-f002:**
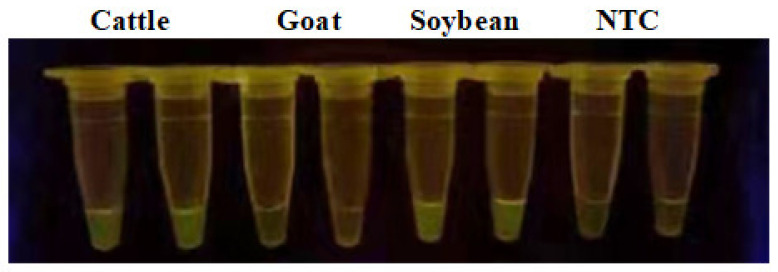
Amplification result based on the best RPA primers for cattle or soybean genomic DNA. NTC: nontarget control.

**Figure 3 foods-13-01637-f003:**
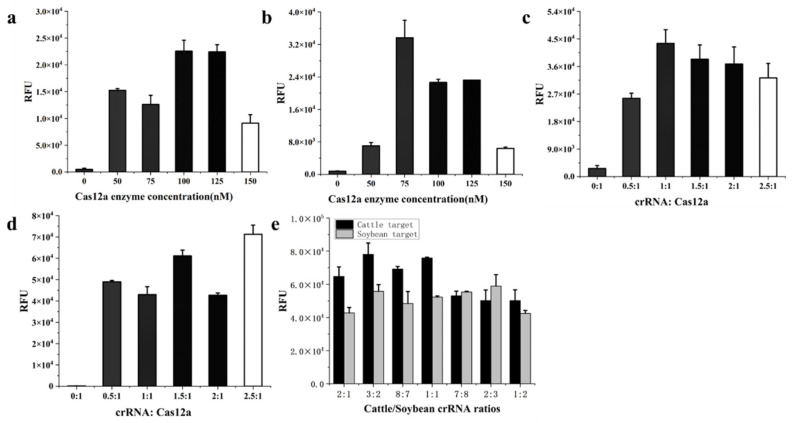
Optimization of dual RPA-CRISPR/Cas12a assay. (**a**) The effect of Cas12a enzyme concentration for cattle target. (**b**) The effect of Cas12a enzyme concentration on soybean target. (**c**) The effect of crRNA/Cas12a ratios for cattle target. (**d**) The effect of crRNA/Cas12a ratios on soybean target. (**e**) Optimization of the ratio of cattle crRNA to soybean crRNA.

**Figure 4 foods-13-01637-f004:**
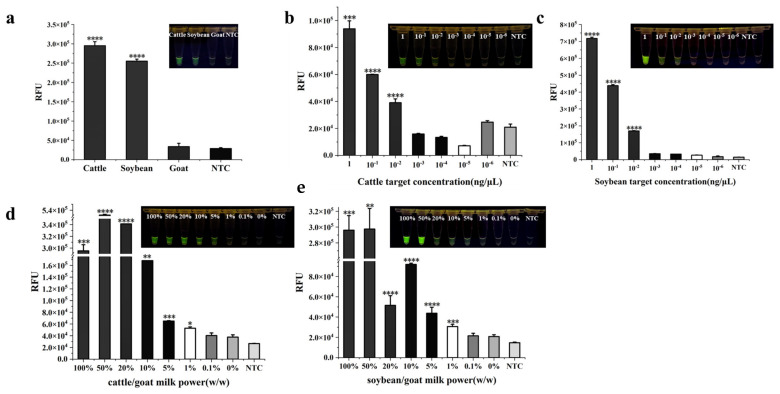
Specificity and detectability of dual RPA-CRISPR/Cas12a. (**a**) The specificity of dual RPA-CRISPR/Cas12a. (**b**) The cattle absolute detectability of dual RPA-CRISPR/Cas12a. (**c**) The soybean absolute detectability of dual RPA-CRISPR/Cas12a. (**d**) The detection limit of cattle-derived adulteration ratio. (**e**) The detection limit of soybean-derived adulteration ratio. NTC: nontarget control. * *p* < 0.05, ** *p* < 0.01, *** *p* < 0.001, **** *p* < 0.0001.

**Figure 5 foods-13-01637-f005:**
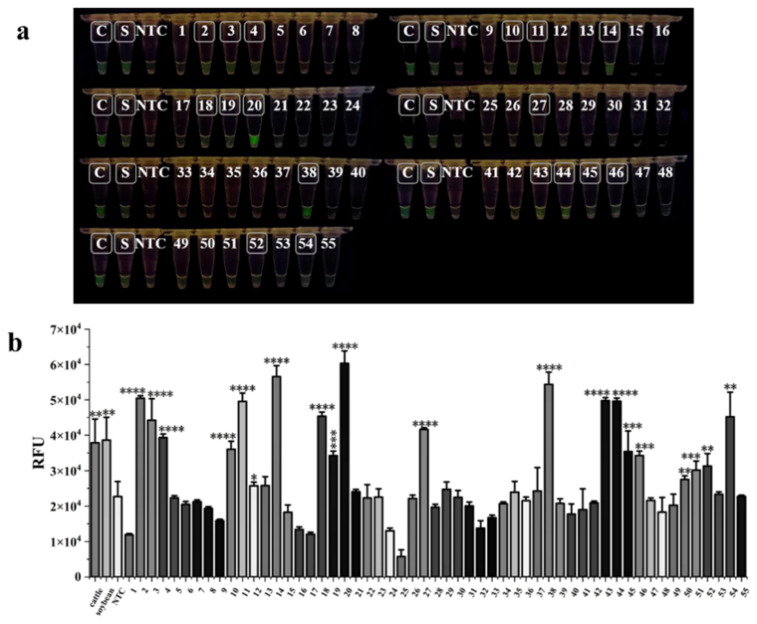
The results of analysis of 55 goat milk powder products tested by dual RPA-CRISPR/Cas12a assay. (**a**) The results of naked-eye observation. The white boxes show that 17 samples exhibited green fluorescence, indicating positive samples. (**b**) The results of data analysis. The unpaired two-tailed Student’s *t*-test was used to analyze the statistical significance of the sample fluorescence values (* *p* < 0.05, ** *p* < 0.01, *** *p* < 0.001, **** *p* < 0.0001). C: positive control of cattle; S: positive control of soybean; NTC: negative control.

## Data Availability

The original contributions presented in the study are included in the article/Appendix A, further inquiries can be directed to the corresponding author.

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
