# Peer review of "A Dual and Rapid RPA-CRISPR/Cas12a Method for Simultaneous Detection of Cattle and Soybean-Derived Adulteration in Goat Milk Powder"

_foods, 2024, doi:10.3390/foods13111637_

Round 1
Reviewer 1 Report
Comments and Suggestions for Authors
Overall comment:
The authors presented a manuscript entitled “A Dual and Rapid RPA-CRISPR/Cas12a Method for Simultaneous Detection of Cattle-Derived and Soybean-Derived Adulteration in Goat Milk Powder” in which a fluorescence method with adulterant DNA interaction, that is cattle and soy milk, where used in order to evaluate adulteration in goat milk. The theme of the article is interesting. The English is average and can use some improvement in many sections. The scientific design of the experiments seems scientifically sound. My major concern in this work, is the poor presentation of the results. The figures have been very poorly made. The results of the commercial samples adulteration analysis are made solely on the visual basis and are not well described, the cutoff for the determination of adulteration is not discussed upon. The authors could easily devise a simple mathematical cutoff (on literature basis) or decision tree model to determine the decision of the adulteration label, however, this is not present in the text. The authors jumped to the conclusion, without clearly describing the final findings and discussing them.
Specific comments:
L234 “salt condition made plenty of milk proteins out of the solution” sentence is vague
L258 “the respective primer concentration was 600 nM” respective to what? again the sentence is somewhat confusing
Figure 2 was poorly made. The authors should consider remaking this figure into an 3x3 figure
Figure 3 should be 2x3 (lines vs. columns)
Figure 4 also has poor quality and arrangement. 2x3 (lines vs. columns). The fluorescence NTC levels inset should be an inset within each subfigure (A to E), also, the label sizes should preferably, but not obligatorily, match between all labels.
Figure 5 should be centered.
L377 “showed that 17 samples produced green fluorescence visible to the naked eye” describe in text exactly which ones, within the text or in table form.
L378-379 “Besides, samples No. 12, No. 50, and No. 51 were also judged positive from the fluorescence values” What does the authors mean by that? It is pretty vague
L381 “were adulterated, accounting for 36.36% of the total samples” which samples? Which samples were adulterated or not? What was the cutoff value in RFU for determining this adulteration? Which reasoning did the authors used to get to that conclusion.
Comments on the Quality of English Language
The English is average and can use some improvement in many sections.
Reviewer 2 Report
Comments and Suggestions for Authors
Dear authors
I carefully checked your paper and the content was intriguing. Please address the following comments while you will revise the manuscript content.
1- Please check your references and make sure problematic papers were not cited within your references.
2- The manuscript text will require a final edition. Please improve the quality of your manuscript English language.
3- The manuscript title is too long. Please select another title for your manuscript.
4- Please provide a list of abbreviations for all summarized terms.
5- In some paragraphs, the manuscript text has different font sizes. Please unify the manuscript text font.
6- Please add a flowchart to the M&M section of your paper to summarize the applied methodology in this paper.
7- Please cite all protocols used in the M&M section.
8- All plots represented in Figure 2 should be moved to the supplementary files.
9- In Figure 5, how the authors assigned **** to the discussed box plots. Please supplement a copy of the raw data used for the statistical analysis.
10- What is the novelty of this study?
11- Further assays will be required to validate the data reported in this paper
Reviewer 3 Report
Comments and Suggestions for Authors
Thank you very much for the opportunity to review the manuscript "A Dual and Rapid RPA-CRISPR/Cas12a Method for Simultaneous Detection of Cattle-Derived and Soybean-Derived Adulter-ation in Goat Milk Powder" .
Please find below some comments.
Please write the bibliography in the text and at the end according to the guidelines of the journal.
Please keep the same font size throughout the text.
Please provide the full name for the acronym CRISPR/Cas.
Line 68 and Line78. What is the difference betwennCRISPR/Cas and CRISPR/Cas12a?
Please rewrite the conclusions emphasizing the main findings of this paper.
Round 2
Reviewer 1 Report
Comments and Suggestions for Authors
The authors answered the specific comments, but did not answered part of my suggestions in the "Overall comments". Please answer all the reviewers comments in your next article. Nevertheless, the specific comments were answered.
The new added parts need an English revision. The English of the manuscript is still not good.
Comments on the Quality of English Language
The new added parts need an English revision. The English of the manuscript is still not good.
Reviewer 2 Report
Comments and Suggestions for Authors
Accept